# Data Factors for Better Compositional Generalization

**Xiang Zhou**  **Yichen Jiang**  **Mohit Bansal**
UNC Chapel Hill
{xzh, yichenj, mbansal}@cs.unc.edu

## Abstract

Recent diagnostic datasets on compositional generalization, such as SCAN (Lake and Baroni, 2018) and COGS (Kim and Linzen, 2020), expose severe problems in models trained from scratch on these datasets. However, in contrast to this poor performance, state-of-the-art models trained on larger and more general datasets show better generalization ability. In this work, to reconcile this inconsistency, we conduct an empirical analysis by training Transformer models on a variety of training sets with different data factors, including dataset scale, pattern complexity, example difficulty, etc. First, we show that increased dataset complexity can lead to better generalization behavior on multiple different generalization challenges. To further understand this improvement, we show two axes of the benefit from more complex datasets: they provide more diverse examples so compositional understanding becomes more effective, and they also prevent ungeneralizable memorization of the examples due to reduced example repetition frequency. Finally, we explore how training examples of different difficulty levels influence generalization differently. On synthetic datasets, simple examples invoke stronger compositionality than hard examples do. On larger-scale real language datasets, while hard examples become more important potentially to ensure decent data coverage, a balanced mixture of simple and hard examples manages to induce the strongest generalizability.[1]

## 1 Introduction

Many recent diagnostic datasets, e.g., SCAN (Lake and Baroni, 2018), COGS (Kim and Linzen, 2020), etc., have exposed the compositional generalization problem of neural sequence-to-sequence (seq2seq) models. They show that seq2seq models that are trained from scratch fail miserably when tested on examples containing novel combinations of seen elements. However, in contrast to these results, models trained (or pretrained) on larger datasets, (including T5 (Raffel et al., 2020), PaLM (Chowdhery et al., 2022), etc.) show substantially better performance (Furrer et al., 2020; Drozdov et al., 2023) for compositional generalization. While some factors behind many of these improvements are the emergent abilities from scaling up (Wei et al., 2022), we explore an alternative and complementary angle: *how and why does training on **more complex** datasets help a general Transformer model, with limited size and capacity, generalize compositionally to unseen natural language queries?*

To answer this question, we empirically study how changes in data factors (e.g., scale, diversity, difficulty, etc.) influence the generalization ability of models trained from scratch. Our first analysis is inspired by Patel et al. (2022) and Jiang et al. (2022), who make the model generalize significantly better on SCAN simply by increasing the number of unique primitives. We extend this observation in several directions and further explore where this gain comes from. Our experiments show that this is not a special observation on SCAN for the *Jump* and *Around Right* split. Instead, the same effect can be observed on multiple different types of generalization challenges (e.g., primitive-level generalization, length-level extrapolation, etc.), and on both synthetic and real-language datasets. To summarize our first direction of analysis, there is a strong connection between *increased dataset complexity* and *better compositional generalization*.

Second, we try to understand *why* more complex datasets lead to better generalization. We build up our analysis by comparing two potentially competing behaviors of the models: *surface memorization* (i.e., memorizing the direct mapping from inputs to outputs without understanding the underlying composition), and *compositional understanding*

---

[1]The code and data for this work are available at https://github.com/owenzx/data4comp.

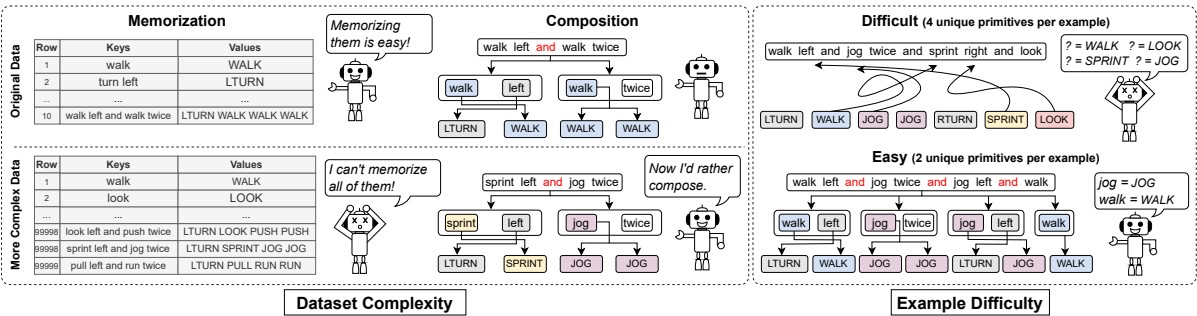

Figure 1: Model's compositional generalization ability is affected by the complexity of the dataset (left) and the difficulty of training examples (right).

(i.e., first discovering a set of compositional rules from examples and then applying the rules for predictions). Please also refer to the left part in Figure 1 for an illustration. While both behaviors can lead to near-perfect training-set performance, only the latter guarantees good generalization performance. We argue that more complex datasets improve compositional generalization as they provide substantial obstacles to surface memorization. Specifically, we make two hypotheses: (1) **diversity**: more diverse patterns (e.g., more unique primitives) exemplified in complex datasets increase the difficulty of surface memorization; (2) **frequency**: the larger dataset size causes a decrease in the frequency of seeing similar examples, thus preventing their memorization. To empirically confirm these hypotheses, we provide detailed ablations showing how both factors contribute to empirical gains in data augmentation. To further isolate the effect of frequently recurring examples, we show that deliberately encouraging example repetition even in a large dataset brings substantial detriment to the generalization performance. Furthermore, as a corollary to our hypotheses, we provide a simple yet effective data augmentation method AugZero that satisfies both contributing factors without utilizing any dataset-specific knowledge.

Finally, we provide a more fine-grained analysis to study how examples of different difficulties inside a dataset affect generalization. Our analysis covers both smaller, synthetic datasets and larger-scale, real-language datasets. On synthetic datasets (e.g., SCAN), we create multiple versions of the datasets strictly controlling the difficulty of the examples in each version, and show that simpler examples facilitate compositional generalization more than difficult examples. On sub-sampling experiments in real-language datasets (e.g., ATIS, SMCalFlow), we observe similar but more complicated trends. Potentially due to the need to cover

all the diverse natural language phenomena, simple examples alone are not enough for achieving good performance, however mixing difficult examples with simple examples is still beneficial on the compositional dataset SMCalFlow-CS.

In conclusion, we present an empirical study on how data factors influence compositional generalization behavior. We notice: (1) increased dataset complexity can improve the model's generalization ability; (2) increased complexity creates obstacles for surface memorization by having a higher cost of memorization and less recurring frequency of examples; (3) example difficulty can influence generalization substantially with simpler examples benefiting compositional generalization more.

## 2 Tasks and Setup

### 2.1 Datasets

In this work, we focus on semantic parsing datasets. We use both datasets with synthetic examples and datasets with natural examples. We experiment with synthetic datasets to have fine-grained control and experiment with flexible natural language datasets to demonstrate the generalizability of our findings. For **synthetic datasets**, we use both the original version of SCAN (Lake and Baroni, 2018) as well as an expanded version with increased complexity, denoted as SCAN* in this work. We introduce SCAN* to control the overall complexity of the SCAN-like dataset.[2] Compared to SCAN, we make two major changes: (1) We remove the constraint that at most one conjunction (*and* or *after*) is allowed for every example. Instead, in SCAN*, every sentence can have an unlimited number of conjunctions. To avoid adding ambiguity brought by multiple conjunctions, we assign higher operation priority to *after* than *and*. (2) We use a larger

---

[2]The original grammar in SCAN does not allow examples with more complexity.

| | Jump | Around Right | GeoQuery (query) | GeoQuery (question) |
|---|---|---|---|---|
| baseline | $3.49_{\pm 1.65}$ | $19.86_{\pm 10.41}$ | $42.37_{\pm 3.26}$ | $64.52_{\pm 1.44}$ |
| 2x Augmentation | $77.37_{\pm 17.74}$ | $73.02_{\pm 20.12}$ | $45.92_{\pm 3.17}$ | $69.10_{\pm 0.85}$ |
| 20x Augmentation | $99.68_{\pm 0.32}$ | $99.38_{\pm 0.68}$ | $47.85_{\pm 3.89}$ | $68.17_{\pm 2.13}$ |
| 200x Augmentation | $99.93_{\pm 0.04}$ | $99.01_{\pm 0.95}$ | $45.70_{\pm 1.66}$ | $65.45_{\pm 2.43}$ |

Table 1: Datasets with increased complexity via data augmentation lead to better compositional generalization. We use logic form outputs for GeoQuery in this table. For SQL results with similar trends, see Table 7 in the Appendix.

set of verb-type primitives (i.e., *run*, *walk*, etc.). The first change allows us to increase the number of possible example structures and lengths, and the second allows the increased lexicon-level complexity. For **natural language datasets**, we use three English datasets with human-written queries: GeoQuery (Zelle and Mooney, 1996), ATIS (Price, 1990; Dahl et al., 1994) and SMCalFlow (Andreas et al., 2020; Yin et al., 2021). For GeoQuery, we use both the *query* split and the *question* split following Andreas (2020). And for SMCalFlow, we include the 32-shot version in Yin et al. (2021) as a compositional split and denote it as SMCalFlow-CS. For fair comparison across splits, we preprocess SMCalFlow and SMCalFlow-CS in the same way. More details are in Appendix A.

## 2.2 Implementation Details

We focus on seq2seq Transformers as they are the most prevalent choices for most semantic parsing datasets. For our main experiments, we train the Transformer from scratch so that our analysis is not influenced by the existing knowledge from pretraining. Our model configuration mainly follows Csordás et al. (2021) to use a 3-layer Transformer which works well in most compositional tasks. We provide additional results on models with other configurations in Appendix H. We decode using a beam size of 5 and select the best model based on their dev-set exact-match accuracy. For any experiments in the same table or figure, we train every model using the same amount of total steps even though the training dataset size may vary. Unless otherwise mentioned, all the results are the mean of 5 runs. More implementation details are in Appendix B.

## 3 Increased Training Set Complexity Leads to Better Generalization.

On the SCAN (Lake and Baroni, 2018) dataset, models trained from scratch show very poor performance, especially on the *Jump* split. However, recent works (Patel et al., 2022; Jiang et al., 2022) use data augmentation methods to significantly im-

prove the performances. Notably, the main effect of these methods is to increase the number of primitives. They neither provide any structure-level change nor create any overlapping between the training and the testing set. Therefore, the improvement seems to be solely from a *more complex* training set. In this section, we replicate and extend their findings, showing this phenomenon is in fact consistent on multiple generalization challenges.

## 3.1 Experiment Design

We first follow the same setup as Jiang et al. (2022) to conduct data augmentation experiments on the SCAN *Jump* and *Around Right* split, creating augmented training sets with more primitives. We also extend this process to GeoQuery dataset (Zelle and Mooney, 1996) to investigate the effect on datasets with more natural and realistic language. Please refer to Appendix C for details of our replication.

Additionally, we include the SCAN* *Length* split to investigate the effect of data complexity on a different type of generalization: length extrapolation. To allow more flexible control of length, we use the expanded SCAN* dataset described in Sec. 2.1. Specifically, we train the model on training sets containing examples with length $0 < l \leq L$, and challenge it on test examples with length $L < l \leq 2L$. To control the complexity, we create four splits with different values of $L \in \{31, 62, 125, 250\}$. For all four splits, we make sure they have similar dataset statistics whenever possible. See Appendix A.2 for more details.

## 3.2 Results

In Table 1, we show the generalization performances when trained with different complexities (i.e., different numbers of total primitives). On both SCAN and GeoQuery, increasing the training-set complexity substantially improves the generalization performance. On SCAN, the models obtain huge improvements with 2x augmentation and reach a near-perfect accuracy with 20x or more primitives. Similarly, on GeoQuery, the augmented performance substantially outperforms the base-

|      | jump        | walk       | look       | run        |
|------|-------------|------------|------------|------------|
| 1x   | -0.84, 0.57 | 0.02, 0.59 | 0.75, 0.15 | 0.20, -0.13 |
| 20x  | 0.41, 0.11  | 0.49, -0.01 | 0.45, 0.50 | 0.75, -0.10 |

Table 2: The projections of "jump, walk, run, look" onto the two most principal components of the embedding space, when trained on *Jump* 1x or *Jump* 20x.

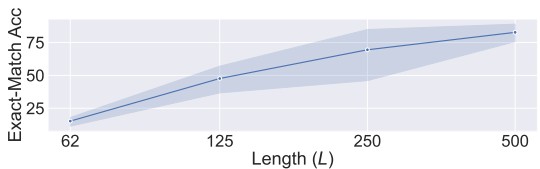

Figure 2: Different generalization performances on SCAN* *Length*. The challenge is always to train on examples with length $0 < l \leq L$ and test on examples with length $L < l \leq 2L$.

line, reaching around 5% accuracy improvement with the best augmentation.[3] Interestingly, there is a difference in the optimal augmentation between SCAN and GeoQuery. On SCAN, larger datasets always lead to better results until the model reaches near-perfect accuracy. However, on GeoQuery, the best augmentation is *not* the most complex 200x data, though a larger amount of augmentation still leads to improvements. We suspect this phenomenon is related to the frequency of primitives, and please see Sec. 4.3 for more discussion.

Besides primitive-level generalization, increased complexity is also beneficial for simple structure-level challenges such as length generalization. In Figure 2, we see steadily increasing performance when both the training set contains longer examples. For the same generalization challenge (i.e., generalizing to twice the length seen during training), the same model can reach over 80% accuracy when the training set contains sentences with 250-token length, but only reaching around 15% if the training sentences are all within a length of 62. These results demonstrate that across multiple different generalization challenges, more complex training sets seem to bring a consistent gain.

**Analysis on learned primitive embeddings.** To better understand how training on a more complex dataset (e.g., *Jump* 20x) changes the model's behavior, we conduct an analysis on the word embeddings of four primitives: "jump, walk, run, look". First, we perform a linear Principal Component

---
[3]Note that unlike SCAN, primitive-level generalization is not the only challenge in GeoQuery, so the scale of the improvement is noticeably smaller. Nonetheless, the improvements shown in Table 1 are consistent across different splits, output formats, and augmentation sizes.

Analysis on the entire embedding matrix and then project these four embeddings to the first two principal components. In Table 2, we show the projection of the embeddings trained on *Jump* 1x versus the embeddings trained on the larger *Jump* 20x. We observe that the four primitives trained on *Jump* 20x are closer together in the first principal component, ranging from 0.41 to 0.75 ($0.53\pm0.13$), compared to -0.84 to 0.75 for the baseline ($0.13\pm0.57$). This suggests that compositionality arises because the model can better represent the syntactic similarity between the rare primitive "jump" and other common primitives, which appear in very different contexts during most of the training.

## 4 Understanding the Advantages Brought by Increased Complexity

We further explore *how* increased complexity improve generalization. We will first provide a discussion on dataset complexity and model behaviors. Then, we will state our hypotheses on how data complexity impacts generalization by influencing the model behaviors. Finally, we provide supporting experiments.

### 4.1 Preliminaries

**Two types of data complexity.** For simplicity, we provide an informal discussion of "data complexity" for semantic parsing tasks. Data complexity can be measured from many different angles. In this study, we only consider the *training set* and mainly focus on two types of data complexity: (1) **Pattern complexity**: a training set with larger pattern complexity will contain more unique patterns (e.g., more unique primitives, more diverse example length, etc.) in the examples. (2) **Scale complexity**: a training set with larger scale complexity will contain more different examples. Note that these two properties are usually positively correlated in most datasets as a larger training set usually contains more diverse patterns. In this section, we aim to explain how increments in these complexities influence the model's generalization.

**Two competitive model behaviors.** One way to understand the different generalization behaviors is to focus on how models achieve good training performance. We argue the difference in how models achieve good training performances heavily influences the generalization performance. Intuitively, for a seq2seq task, there are two extremes of behaviors that models could adopt: (1) **surface mem-**

**orization**: the model only establishes one-to-one maps from the inputs and the outputs and does not correctly infer the composition;[4] (2) **compositional understanding**: the model makes predictions based on a correct understanding of how the semantics are composed by smaller sub-structures.[5] An illustration of these two behaviors is shown in Figure 6 in the Appendix. In practice, the model's behavior is not a binary choice between these two extremes, but oftentimes a complicated combination depending on the inputs. Nonetheless, here we use these two concepts to denote two trends of behaviors, and we will show that increased data complexity biases the model toward the second behavior, and hence leads to better compositional generalization.

### 4.2 Hypotheses

We elaborate on our hypotheses of the connection between data complexity and model behaviors. Note that our two hypotheses are interrelated and not contradictory to each other. They are just two different perspectives on how this connection.

**(1) Pattern complexity (i.e., more diverse examples) increases the difficulty of surface memorization.** For surface memorization, the models need to memorize all the individual mappings from each example input to each example output. The difficulty of achieving a specific training loss through surface memorization is proportional to the total sum of the complexity of *every* unique example in the dataset. With a larger training set containing more diverse examples, the difficulty of surface memorization increases substantially. However, with correct compositional understanding, the difficulty will remain roughly the same as the previously-learned composition remains correct. Hence, with more complex datasets containing more different examples, the difficulty of surface memorization increases much faster than the

---

[4]This may remind readers of classic *overfitting* behaviors. Indeed, traditional loss regularization methods (Li et al., 2019; Yin et al., 2023) can improve compositional generalization to an extent. However, finding the best regularization method is not trivial, and we also need to understand methods (e.g., LLMs) free of explicit regularization. Our work aims to provide related insights from the dataset angle.

[5]Depending on the actual dataset properties, compositional understanding may not always be the correct behavior (e.g., in translation as shown in Dankers et al. (2022)). However, for the scope of this study, we focus on how more complex datasets encourage compositional generalization, so we assume compositional understanding as the ideal behavior.

| Dataset | SCAN (*Around Right*) | GeoQuery (query) |
|---|---|---|
| Origin Dataset | $19.86_{\pm 10.41}$ | $42.37_{\pm 3.26}$ |
| 2x Augmentation | $73.02_{\pm 20.12}$ | $45.92_{\pm 3.17}$ |
| + More primitives | $96.09_{\pm 4.33}$ | $43.55_{\pm 1.82}$ |
| + More prim. & larger size | $99.38_{\pm 0.68}$ | $47.85_{\pm 3.89}$ |
| + AugZero | $95.50_{\pm 7.48}$ | $43.55_{\pm 1.01}$ |

Table 3: Both dataset size and the number of primitives contribute to the performance improvement.

difficulty of compositional understanding, leading to a preference for the latter.

**(2) Scale complexity (i.e., larger datasets) avoids memorization by reducing frequently recurring examples.** Another effect usually brought by increased dataset complexity is more training examples. This naturally leads to a decreased frequency with which similar examples occur during the training. As is shown by Hernandez et al. (2022), even a small portion of recurring data can bring substantial damage to the copying performance and the scaling law of a language model. We argue that similar to this finding, reduced example recurring frequency will make surface memorization harder and thus encourage compositional understanding.

### 4.3 Supporting Experiments

In this section, we provide empirical evidence supporting our previous two hypotheses.

**Two sources of empirical improvements.** First, we try to separate the benefits of data augmentation shown in Sec 3 into two parts: benefits from increasing pattern complexity and from increasing scale complexity. In Table 3, we present models trained with data of three different level of complexities. The "+ More prim. & larger size" row is the model trained with the full x20 data augmentation with more primitives *and* a larger dataset size. For the "+ More primitives" row, we down-sample the augmented part of the x20 data[6] so that the total size is the same as the x2 augmented dataset.[7] On both SCAN and GeoQuery, by comparing 'Origin Dataset' and '+ More primitives' rows, we find that increasing pattern complexity can encourage stronger compositional generalization. Additionally, by comparing the '2x Augmentation' and the '+ More primitives' row, we notice that the trade-off of having more primitives but fewer examples

---

[6]We do not down-sample the whole dataset as it will create an unwanted side-effect of removing a large portion of the original data compared to the x2 data.

[7]Note that there is no ablation *only* with larger sizes, since if we keep the original examples unchanged, increasing dataset size will inevitably require new primitives.

| Data Size | Repetition | GeoQuery (query) | GeoQuery (question) |
|---|---|---|---|
| Original | / | $42.37_{\pm 3.26}$ | $64.52_{\pm 1.44}$ |
| Original | Example | $24.84_{\pm 3.64}$ | $55.48_{\pm 4.51}$ |
| Original | Primitive | $1.65_{\pm 1.69}$ | $43.08_{\pm 0.93}$ |
| 20x | / | $47.85_{\pm 3.89}$ | $68.17_{\pm 2.13}$ |
| 20x | Example | $44.95_{\pm 3.20}$ | $68.46_{\pm 1.13}$ |
| 20x | Primitive | $31.40_{\pm 4.82}$ | $49.11_{\pm 0.84}$ |

Table 4: The performance of models trained with different types of example repetition curriculum.

for each primitive is not always beneficial, leading to improvement on SCAN, but not on GeoQuery. Finally, by comparing '+ More primitives' and '+ More prim. & larger size', we can further conclude that increasing scale complexity given the same amount of distinct primitives can provide further gain in generalization results.

**Performance detriment from example repetition.** To further support our second hypothesis, we present experiments to show that highly-recurring examples can cause a significant performance decrease. We compare three training curricula with the same total steps in Table 4. The first curriculum with no repetition is a normal curriculum where every epoch contains the entire training set. The other two curricula are specifically designed to aid surface memorization by having more recurring examples. For the Example Repetition curriculum, at the first 20% of the training steps, the model is only trained on a small subset containing 20% of the examples, then the remaining data are gradually put back into the training set until the training set becomes the full dataset at 80% of the total steps. This curriculum ensures that the model is repeatedly trained on a small set of examples for a long period. The Primitive Repetition curriculum is similar to the Example Repetition curriculum except that it first clusters the training examples by the primitives and then starts to train the model with data containing only 20% of the primitives. In Table 4, we confirm that example repetition can bring substantial damage to the generalization performance. On smaller original datasets, repetition at both the example level and the primitive level substantially hurt the performance. On GeoQuery, Example Repetition leads to a 10 to 20 accuracy drop, and the damage brought by Primitive Repetition is even larger. With a larger dataset resulting from data augmentation, the trend is slightly different. Repetition at the example level causes a much smaller drop, possibly because that 20% of the augmented dataset is still relatively large so there is not much repetition. However, primitive-level repe-

tition still causes substantial damage, showing over 15 points drop. To summarize, earlier in this paper, we show that increasing the dataset size generally helps generalization. However, Table 4 show that naturally increasing the dataset size while still explicitly adding highly recurring examples actually leads to worse results. Combining these two observations, we provide evidence supporting our hypothesis on the importance of reducing frequently recurring examples.

**Simple Data Augmentation with *Zero* Prior Knowledge** Finally, we provide a data augmentation method as a direct corollary of our hypotheses. We call our method *Augmentation with Zero Prior Knowledge* (AugZero) as it brings zero additional knowledge but can still be effective as it satisfies our two previous two hypotheses. The main idea is to simply copy the *entire* vocabulary and use the newly copied vocabulary to re-tokenize the entire dataset. Figure 3 shows an illustration for AugZero with $k$ times augmentation. A more detailed description is in Appendix C.3. The results with $k = 200$ are in Table 3. Despite how simple the algorithm is, AugZero can still substantially improve the performance, reaching close to perfect performance on the SCAN *Around Right* split and achieving decent improvement on GeoQuery. While the gain is smaller than the primitive-aware augmentation result (+ More prim. & larger size row), its success further supports our hypotheses.

## 5 Which Examples Benefit Generalization the Most: A Difficulty Perspective

Previously, we demonstrate and explain that increasing the complexity of the whole dataset can improve compositional generalization. In practice, examples *inside* a dataset have different properties (e.g., difficulty, topic, etc.) and thus may influence generalization differently. In this section, we present a study from the difficulty perspective. We will start with a discussion on the definition of difficulty, and then show how example-level difficulty can substantially influence compositional generalization on both synthetic and real datasets. For this section, we report results with three runs as the dataset sizes are substantially larger.

### 5.1 Example-Level Difficulty Metrics

**Complexity-based difficulty.** For synthetic datasets like SCAN, we can intuitively define the difficulty of an example by the complexity of the

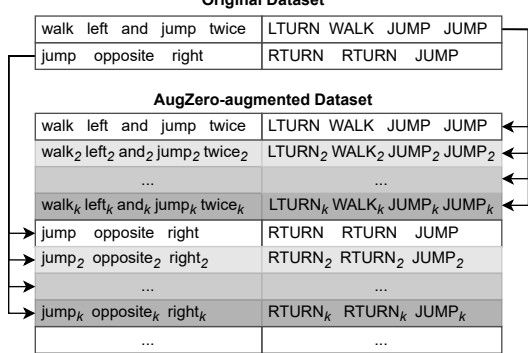

**Original Dataset**

| walk left and jump twice | LTURN WALK JUMP JUMP |
| jump opposite right | RTURN RTURN JUMP |

**AugZero-augmented Dataset**

| walk left and jump twice | LTURN WALK JUMP JUMP |
| $walk_2$ $left_2$ $and_2$ $jump_2$ $twice_2$ | $LTURN_2$ $WALK_2$ $JUMP_2$ $JUMP_2$ |
| ... | ... |
| $walk_k$ $left_k$ $and_k$ $jump_k$ $twice_k$ | $LTURN_k$ $WALK_k$ $JUMP_k$ $JUMP_k$ |
| jump opposite right | RTURN RTURN JUMP |
| $jump_2$ $opposite_2$ $right_2$ | $RTURN_2$ $RTURN_2$ $JUMP_2$ |
| ... | ... |
| $jump_k$ $opposite_k$ $right_k$ | $RTURN_k$ $RTURN_k$ $JUMP_k$ |
| ... | ... |

Figure 3: The AugZero data augmentation process.

example itself, for example (1) the length of the input instruction, or (2) the number of unique primitives, both reflecting the pattern complexity of the example. These metrics are the most reliable as they directly reflect the complexity of correctly generating the target output examples. However, these metrics can be hard to measure on real natural language datasets.

**Prototype-based difficulty.** Another limitation of the complex-based difficulty is that it treats each example independently, while in practice the learning is conducted on an entire dataset instead of individual examples. One metric addressing this issue is to see how prototypical (i.e., if there are other similar examples in the dataset) one example is. To measure this, we follow the process in Sorscher et al. (2022). We use SimCSE (Gao et al., 2021) to encode all the input of the examples, then cluster the encoded vectors using K-means, and use the L2 distance to the centroid as the difficulty measure. An outlier that is far from any cluster is deemed difficult. See Appendix G.1 for clustering examples for this method. In our experiments, we will use this difficulty as the metric when the ground truth complexity metric is not available.[8]

### 5.2 Experiments

We show how models behave differently with different distributions of example difficulties.

**Simpler examples make generalization easier on SCAN*.** First, as we have full control over SCAN*, we directly use example complexity (including both length-based and primitive-based

---

| Datasets | Simple | Hard | Mix |
|---|---|---|---|
| ATIS | $40.86_{\pm0.43}$ | $52.26_{\pm1.05}$ | $48.11_{\pm0.94}$ |
| SMCalFlow | $37.35_{\pm0.26}$ | $46.38_{\pm0.55}$ | $43.69_{\pm0.35}$ |
| SMCalFlow-CS | $36.72_{\pm0.70}$ | $38.43_{\pm1.03}$ | $39.41_{\pm1.11}$ |

Table 5: Different performance when trained subsets with different difficulty. Simple and Hard denotes the simplest and hardest subset as in Figure 4. Mix is a mixture of both subsets.

complexity) as the difficulty metric. For evaluation, we use the *Jump* split as its difficulty is substantially influenced by both types of complexity. For length-based complexity, we generate training sets with different maximum lengths, the same as described in Sec. 3. For primitive-based complexity, we keep the number of total possible primitives in the vocabulary fixed but generate multiple different training sets only varying the maximum number of unique primitives *per example*. Intuitively, it will be easier to infer the correct composition from shorter examples or examples containing fewer primitives. The results are shown in Fig. 5. All the models are tested on the same testing set similar to the original SCAN *Jump* testing set. For both settings, we can see a steadily decreasing trend when the examples become harder. When the maximum length is reduced from 500 to 62, the performance is increased from 16.65% to 47.31%.[9] With only 2 unique primitives per example, the performance is also increased to 49.14%. These results demonstrate that easier examples can make the correct composition easier to learn.

**Mix of simple and hard examples needed on real language datasets.** We next examine the impact of example difficulty on more complex larger-scale natural language datasets. Due to the flexible and diverse nature of natural language in real datasets, models now not only need to understand the correct composition but also need to capture other language uses through potentially non-compositional ways. Therefore, the trends in natural language datasets can be different from the previous observation. For this study, we conduct experiments on ATIS (Price, 1990; Dahl et al., 1994), SMCalFlow (Andreas et al., 2020) and the compositional version SMCalFlow-CS (Yin et al., 2021). For all three datasets, we train models on multiple subsets with different difficulties but all con-

---

[8]Another way to measure difficulty is to directly use the model's performance (e.g., accuracy). Using these metrics, models will perform very poorly when trained on the hardest subset as they fail to learn from most examples, making the analysis results less interesting. See Appendix G.2 for more discussion on these metrics and detailed results.

[9]Note that the results here and the results in Fig. 2 are not contradictory. Here we show more examples with longer lengths make primitive-level generalization worse, while in Fig. 2 we show such data make length generalization better.

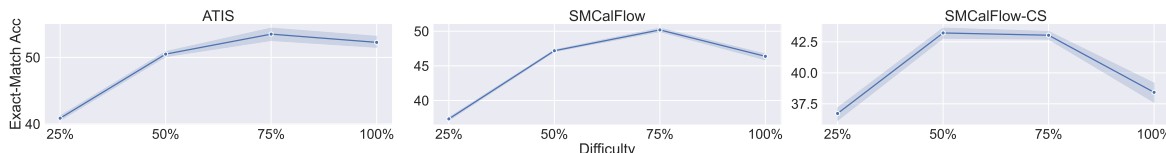

Figure 4: Results on ATIS and SMCalFlow with training sets of different difficulties. The X-axis represents the quantiles of example difficulty, the smaller the easier.

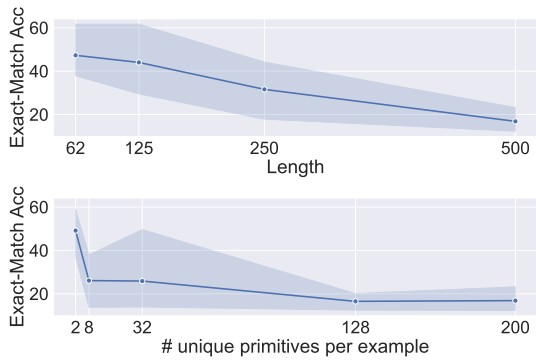

Figure 5: On SCAN* *Jump* split, simpler examples facilitate better compositional generalization.

tain 25% of the training examples. Since on these datasets, we no longer have access to the ground truth example complexity, so we present results with prototype-based difficulty in Figure 4.

We observe similar but more complicated trends on larger-scale natural language datasets. First, we observe that *difficult examples become important on these datasets*. On all three datasets, using only the easiest examples leads to the worst results. We suspect that only using the simplest examples does not provide enough coverage for all the diverse linguistic phenomena in the dataset. However, the best performance is also not achieved with the most difficult examples, but at the medium level. Additionally, we notice that *even in larger-scale natural language datasets, simpler examples are still important for compositional generalization.* Here we specifically focus on the trend difference between the results on SMCalFlow-CS (a compositional split), and the other two non-compositional-split datasets. In Figure 4, we see SMCalFlow-CS achieves the best performance with the second easiest split, different from the other two datasets. Additionally, in Table 5, we show the performance by mixing the hardest examples with the easiest examples with a 50%/50% ratio, and we also observe a unique trend on SMCalFlow-CS. Only on SMCalFlow-CS, the mixed performance outperforms the performance by only using the difficult data, showing the advantage of having simpler ex-

amples for compositional challenges still persists.

## 6 Related Work

**Compositional generalization.** Earlier related works in compositional generalization study the systematic behavior of neural networks in language learning (Wong and Wang, 2007; Brakel and Frank, 2009), compositional counting (Wiles, 1998; Weiss et al., 2018), syntax learning (Linzen et al., 2016), etc. Recently, many recent datasets (Lake and Baroni, 2018; Kim and Linzen, 2020; Loula et al., 2018; Bastings et al., 2018; Keysers et al., 2020; Tsarkov et al., 2021; Hupkes et al., 2020) substantially facilitates the development and evaluation of related method improvements. Since then, many different methods have been proposed, including architecture improvements (Dessì and Baroni, 2019; Gordon et al., 2020; Oren et al., 2020; Zheng and Lapata, 2021), grammar-based approaches (Shaw et al., 2021; Kim, 2021), task decomposition (Herzig et al., 2021), data augmentation (Andreas, 2020; Akyürek et al., 2021; Akyurek and Andreas, 2023), and novel learning methods (Jiang and Bansal, 2021; Lake, 2019; Conklin et al., 2021; Jiang et al., 2022). Of all these methods, the biggest breakthrough (Drozdov et al., 2023; Qiu et al., 2022) still mainly comes from using large language models (Chowdhery et al., 2022; Raffel et al., 2020). Our work aims to provide insight into how (pre-) training on a large corpus is beneficial for compositional generalization. Besides data factors, the generalization behavior of models can also be studied from many different angles, including architecture (Li et al., 2019), regularization (Yin et al., 2023), representation geometry (Montero et al., 2021; Ito et al., 2022; Murty et al., 2023), etc. All these directions are complementary towards the same goal of understanding the mechanism of generalization.

**Dataset influence on generalization.** Prior to the development of LLMs, most dataset quality-related research focuses on dataset artifacts (Gururangan et al., 2018), or label quality (Maynez et al., 2020;

Pavlick and Kwiatkowski, 2019). After the recent success of LLMs, a number of works pointed out that large-scale datasets can be a major contributing factor to the success. Chan et al. (2022) shows that the scale of datasets is as important as the scale of the model. Hoffmann et al. (2022) show how example distribution can significantly impact the emergence of in-context few-shot learning. Hernandez et al. (2022) notice that even a small portion of repeated data can have a significant impact on the generalization performance.

## 7 Discussion

**Advantage of large-scale pretraining.** In this work, we have demonstrated how different data factors (e.g., scale, diversity, example difficulty, etc.) can substantially improve the model's ability to generalize compositionally. While our experiments are done on a smaller scale, these results also hint at why models pretrained on larger datasets show better generalization ability. We argue that doing large-scale pretraining implicitly satisfies many beneficial data factors studied in this work. During pretraining, models are exposed to a large set of different structures and primitives. Additionally, pretraining datasets have a large size that prevents frequent repetition and contains a diverse set of simple and difficult examples. All these conditions incentives the emergence of compositional generalization ability.

**How about fine-tuning models?** In our main experiments, we choose to not use pretrained models as they already possess a substantial amount of knowledge, which makes it very difficult to evaluate how models initially acquire compositionality. Nonetheless, as fine-tuning models are of great practical importance, here we discuss related issues about fine-tuning models and summarize our preliminary findings with T5 models (more details are in Appendix H.2). The behaviors of fine-tuning models are slightly different. We notice that the size and the diversity of the dataset can still be important as repetitive training on similar examples (e.g., with the same primitive) will cause substantial overfitting and hurt the pretrained model. However, as pretrained models already have a good understanding of basic language compositionality, they may need less assistance from the data, so maintaining a large number of different primitives or simple examples may be less important. Overall, our results suggest that when fine-tuning on

small datasets, data augmentation methods increasing the size and diversity may help the model's compositional generalization performance.

## 8 Conclusion

We empirically study how dataset factors influence compositional generalization. We show that increased dataset complexity facilitates compositional generalization, with dataset size and pattern complexity as two important factors behind the gain. We also provide an analysis of example difficulty and discuss the implication of our work on pertaining and fine-tuning models.

## Limitations

The analyses in this study focused on relatively small-scale Transformer seq2seq models. While such architectural choice has been shown to be effective on compositional generalization datasets when trained from scratch (Csordás et al., 2021), very large models may demonstrate different behaviors due to their *emergence abilities* (Wei et al., 2022). The data factor analysis is not meant to be a complete investigation about *all* possible data factors. In this paper, we focus on scale, difficulty, and data complexity, which are all shown to be important factors for compositional generalization. However, there are other factors also being important for compositional generalization. The specific evaluation setup of compositional generalization also varies in different works. In this work, we focus on synthetic diagnostic datasets ensuring sufficient supervision and larger-scale natural language datasets with minimal manual control of the data distribution. Additionally, compositional generalization challenges include both lexicon-level generalization and structure-level generalization. This work mainly focuses on lexicon-level generalization and relatively simple structure-level generalization as in our length-related and SMCalFlow-CS experiments. We assume these results will also generalize to more complicated structure-level generalization problems. For example, some more challenging setups may involve splits maximizing the compound divergence (MCD) (Keysers et al., 2020). MCD complexity does not naturally change when collecting larger datasets. To test a related hypothesis, one may check if increasing the number of unique local structures improves performance. However, such dataset modification is non-trivial and we leave the exploration as future work.

## Acknowledgements

We thank Dipanjan Das, Colin Raffel, and the reviewers for their helpful comments. This work was supported by ONR Grant N00014-18-1-2871, NSF-CAREER Award 1846185, and DARPA MCS Grant N66001-19-2-4031, and an Apple PhD Fellowship. The views are those of the authors and not of the funding agency.

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

# Appendix

# A  Dataset Details

## A.1  Natural Dataset Details

We use three English datasets with human-written queries: GeoQuery (Zelle and Mooney, 1996), ATIS (Price, 1990; Dahl et al., 1994) and SMCalFlow (Andreas et al., 2020; Yin et al., 2021).

**GeoQuery**  (Zelle and Mooney, 1996) dataset contains 880 questions about US geography with the corresponding SQL queries or logical expressions of the question. Following Andreas (2020), we report performance on both the *query* split and the *question* split. The *question* split is the original split proposed in the dataset, while the *query* is a more compositionally challenging split proposed in Finegan-Dollak et al. (2018) ensuring no overlapping logical forms between the splits. We also use the standard dev-test split for GeoQuery.

**ATIS**  (Price, 1990; Dahl et al., 1994) is a larger-scale semantic parsing dataset with 3809 examples. Each example contains a flight-related query as well as the corresponding SQL query. Due to the nesting format difference between the train/dev set and the test set of ATIS, we simplify our experiment setting to report dev set results only for the ATIS dataset.

**SMCalFlow**  (Andreas et al., 2020) is a large-scale dialog dataset. All the examples in the dataset are from natural conversations and they are annotated with an executable dataflow program. Yin et al. (2021) proposed a modified version of this dataset focusing on compositional generalization. We use the 32-shot version in Yin et al. (2021) and denote it as SMCalFlow-CS in later experiments. As our experiments also aim to provide a fair comparison between the original version and the compositional version of SMCalFlow, we pre-process both versions using the same way as the preprocessing steps for the compositional version

described in Yin et al. (2021). Due to this preprocessing detail, our result on the non-compositional split of SMCalFlow may not be comparable to other works.

## A.2 Dataset Construction Details for SCAN* Length Generalization Experiment

SCAN* is an extended version of SCAN created in this work so that we can analyze how dataset complexity (especially length-related complexity) influences the model's generalization ability. This is an essential design choice as the complexity of the original SCAN is bounded. For all SCAN* related experiments, we apply two changes: first, every dataset will contain 200 different primitives; second, every sentence can have an unlimited number of *and* and *after* as long as the total length is within the limit for the length generalization experiments. To avoid ambiguity, we assign different orders of priority to *and* and *after*, with *after* having the higher priority and will be executed first. Additionally, we make two minor simplifications to the SCAN grammar that do not influence the trends substantially. We removed the verb *turn* and we unify the grammar for *around* and *opposite*[10]. To generate the data for the length generalization experiment in Sec. 3, and the length ablation experiment in Sec. 5, we try to ensure similar dataset statistics across datasets with different lengths as much as possible. To achieve this, we always first generate the dataset with the maximum length. Then, to create all the other shorter experiments, we repeatedly create a half-length truncated version of the longer original dataset. Specifically, for every example with length $l$ in the longer dataset, we first split each example by its conjunctions (*and* or *after*), then we keep the most amount of the split parts so that the total length is under the half-length $l/2$ of the original example. For our experiments, we use the length of the input as the length measure.

## B Implementation Details

### B.1 Models and Training Details

In this study, we focus on seq2seq Transformers as they are the most prevalent choices for most compositional generalization (and more broadly, semantic parsing) datasets. For our main experiments, we focus on models trained from scratch so that our analysis is not influenced by the existing

knowledge from pretraining. For the experiments in the main paper, we follow Csordás et al. (2021) to use 3 layers in both encoder and decoder, and use relative positional embeddings (Shaw et al., 2018). In Appendix H, we provide additional results on models with other configurations. We set both the hidden size and the embedding size to 256, and the dimension of the feed-forward layer to 512. We use a dropout rate of 0.1 and 4 self-attention heads. For the optimizer, we use Adam (Kingma and Ba, 2015) and a batch size of 128 examples. We use the Noam Learning rate scheduling with a peak learning rate of 2.0, 50000 total steps, and 5000 warmup steps. We use a beam size of 5 to get all the results in our experiments.

### B.2 Implementation of Difficulty Metrics

For the prototype-based difficulty metric, we follow the process in Sorscher et al. (2022). For every example, we feed the input side to the SimCSE (Gao et al., 2021) model to get an example embedding. We choose to use the input side instead of the output side to compute the embedding since the inputs are natural language so they are more suitable for off-the-shelf models like SimCSE. Then, we run k-means to get the distance between each example embedding and its corresponding cluster centroid. Our k-means implementation is from (Pedregosa et al., 2011). We set k to 100 in our experiments, and use the best clustering result from 10 different initializations.

For the learning learning-based difficulty metric, we follow our standard setup to evaluate our model on the training set for every 500 steps. We then log the performance of each example. Then for every example, we find the earliest step when the model has predicted the example correctly for 10 consecutive steps. The earlier this step is, the simpler this example is. If the model can never predict the example correctly, this step number is set to the final step. In our experiment, we compute this metric on multiple random seeds. Therefore, for every example, the final step number is the average over all the different seeds.

## C Data Augmentation Details

### C.1 Data Augmentation Details of SCAN *Jump* and *Around Right*

We first follow the exact same setup in Jiang et al. (2022) to conduct data augmentation experiments on the SCAN *Jump* and *Around Right* split to create

---

[10]In Appendix E, we show this simplification does not influence the trends shown in this paper

training sets with different numbers of primitives. The augmentation follows a two-stage procedure, which we explain here.

**Building a Dictionary.** In the first stage, we use a dataset-agnostic, rule-based algorithm to build a dictionary that maps certain input tokens to their output forms (e.g., "*look* $\mapsto$ LOOK" and "*jump* $\mapsto$ JUMP"). Given the source vocabulary as $V$ and the target vocabulary as $W$, the algorithm iterates through the training set to identify pairs $(v, w), v \in V, w \in W$ such that the presence of $v$ in the input is both *necessary* and *sufficient* for the presence of $w$ in the output.

$$\text{suff}(v, w) = \forall (x, y), (v \in x) \rightarrow (w \in y)$$
$$\text{ness}(v, w) = \forall (x, y), (w \in y) \rightarrow (v \in x) \quad (1)$$

This algorithm ignores those functional words (e.g., "*around*" and "*twice*") that only decide the syntactic structure of the outputs but cannot be translated to a specific target token.

**Mutating Primitives.** In stage two, we iterate through every example in the training set and randomly select some primitive pairs that exist in the previously built lexicon for mutation. In mutating them, we simply add a suffix to their source and target forms. Given an original example "*walk left twice*", we select a primitive "*walk*" and mutate it to a new example "***walk1** left twice* $\mapsto$ TL **WALK1** TL **WALK1**". For our experiments in Sec. 3, we create three different augmented training sets (2x, 20x, 200x) with increasing complexities. Specifically, for a training example in the SCAN *Jump* or *Around Right*, we first identify all primitive pairs in the example. Then, in $K$x augmentation ($K$=2,20,200), for each primitive pair (e.g., "(*walk, WALK*)"), we randomly replace the source token in the input with one of its $K + 1$ mutated form (*walk, walk1, walk2, ..., walkK*) and the target token in the output with the corresponding form. We repeat this process $2K$ times or until we collect $K$ distinct augmented examples.

## C.2 Data Augmentation Details for GeoQuery

Here we describe the detailed data augmentation process for the GeoQuery dataset as used in Sec. 3 and Sec. 4. As one crucial part of the questions in GeoQuery are geography entities (e.g., *Oregon, Springfield*, etc.), our augmentation focuses on increasing the diversity of these entities similar to the primitive-based augmentation method in Patel et al. (2022) and Jiang et al. (2022). Specifically, we first

get all the geography entities in the dataset by parsing the output predictions. Then, for each entity, we create multiple copies (e.g., 19 new copies for the x20 experiments) for the entity. Then, for every example, we identify the entities contained in every example, and then augment new examples containing the new copies of these entities. We augment the same amount of new examples for every example. The final size of the x20 augmented dataset will be 20 times of the original dataset.

## C.3 AugZero Augmentation Details

Below we describe *Augmentation with Zero Prior Knowledge* (AugZero). The general idea is just to satisfy our hypotheses by providing multiple copies of the training set using different copies of the vocabulary. Interestingly, this process requires *zero* knowledge about the actual data. Specifically, assume the original vocabulary is $V^1 = \{w_j^1 |_{j=1}^m\}$, where $w_1^1 \ldots w_m^1$ are all the $m$ possible tokens. AugZero increases the vocabulary by $k$ times. The augmented vocabulary will be $V^{ADV} = V^1 \cup V^2 \cup \ldots \cup V^k = \{w_j^i : |_{j=1}^m {}_{i=1}^k\}$, where each $w_j^i$ is a corresponding new token for $w_j^1$. Given a tokenized example with length $l$ in the original dataset: $x = w_{u_1}^1, w_{u_2}^1, \ldots, w_{u_l}^1$, where $u_1, \ldots, u_l$ are the token indexes in $V^1$. In AugZero, we create $k - 1$ additional copies of this example, from $w_{u_1}^2, \ldots, w_{u_l}^2$ to $w_{u_1}^k, \ldots, w_{u_l}^k$. See Figure 3 for an illustration. One crucial difference between AugZero and the data augmentation described in Sec. 3 is that AugZero no longer distinguishes the primitive and non-primitive tokens, hence requiring zero knowledge about the actual task.

**Other variants of AugZero.** In AugZero, we select a different set of vocabulary for each example. For example, the original "*walk left and jump*" may become "*walk2 left2 and2 jump2*". If we combine this idea and syntax induction methods, we can get additional augmentation examples such as "walk left2 and2 jump2". This can further improve the performance (as shown by the prim2primX results using very similar ideas). However, doing such augmentation requires inducing the syntax mapping beforehand, which is not trivial on natural language datasets and loses the advantage of simplicity. Additionally, we explored another variant that maps each original token in the dataset randomly into all the corresponding words in the augmented vocabulary. This makes the one-to-one map "*walk* $\mapsto$ WALK" becomes a many-to-many map "{*walk*,

| Dataset Size | Repetition | GeoQuery (query) | GeoQuery (question) |
|---|---|---|---|
| baseline | / | $29.67_{\pm 5.31}$ | $60.79_{\pm 1.69}$ |
| baseline | Example | $8.68_{\pm 4.17}$ | $56.63_{\pm 2.50}$ |
| baseline | Primitive | $13.87_{\pm 2.70}$ | $36.77_{\pm 1.90}$ |
| 20x | / | $32.97_{\pm 2.60}$ | $62.44_{\pm 1.80}$ |
| 20x | Example | $30.00_{\pm 1.85}$ | $62.65_{\pm 1.38}$ |
| 20x | Primitive | $6.48_{\pm 2.21}$ | $40.05_{\pm 1.07}$ |

Table 6: The effect of examples repetition on different datasets with SQL outputs. We use SQL outputs for the GeoQuery dataset in this table.

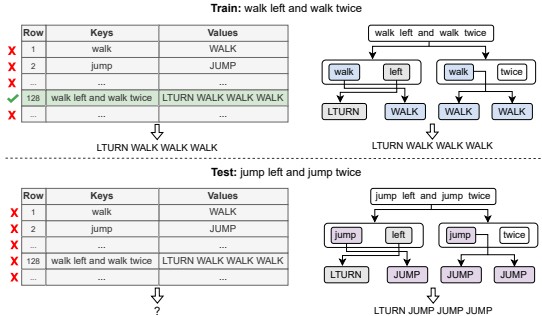

Figure 6: Two different ways to achieve low training losses: surface memorization (left) and compositional understanding (right).

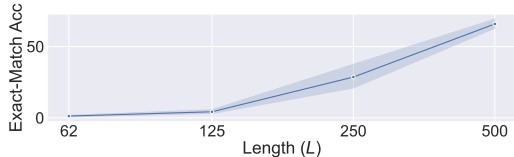

Figure 7: Different generalization performances on without the grammar simplification of *around* and *opposite*. Same to Figure 2, the challenge is always to train on examples with length $0 < l \leq L$ and test on examples with length $L < l \leq 2L$.

*walk1*, *walk2*, ...} $\mapsto$ {WALK, WALK1, WALK2, ...}", but do not show further improvements.

## D Illustration for the Two Model Behaviors

In Figure 6, we provide an illustration for the two model behaviors mentioned in Sec. 4.1. Both behaviors can lead to good performance on the training set (as shown in the upper half of the figure). However, given a new example with a novel combination of seen structures and primitives, only models with correct compositional understanding can provide the correct prediction (as shown in the lower half of the figure).

|  | GeoQuery (query) | GeoQuery (question) |
|---|---|---|
| baseline | $29.67_{\pm 5.31}$ | $60.79_{\pm 1.69}$ |
| 2x Augmentation | $26.05_{\pm 4.34}$ | $61.72_{\pm 2.35}$ |
| 20x Augmentation | $32.97_{\pm 2.60}$ | $62.44_{\pm 1.80}$ |
| 200x Augmentation | $32.64_{\pm 1.00}$ | $61.08_{\pm 2.11}$ |

Table 7: Datasets with increased complexity via data augmentation are easier for compositional generalization. We use SQL outputs for the GeoQuery dataset in this table.

## E Additional Length Generalization Results

In Figure 7, we show length generalization results without the grammar unification of *around* and *opposite*. We can see the trend is very similar to Figure 2. As the maximum length becomes larger in the dataset, generalization performance becomes better. In our preliminary experiments, we have verified that our other findings on SCAN* are also robust against minor grammar changes.

## F Additional GeoQuery Results

In the main paper, due to space constraints, we report on the GeoQuery dataset with logical forms as output. In Table 6 and Table 7, we show the corresponding results using SQL as the output. Both tables demonstrate similar trends to the results in the main paper.

## G Additional Example Difficulty Results

### G.1 Examples of Prototype-Based Difficulty

In Table 8, we show examples of simple and difficult data in the SMCalFlow-CS dataset according to the prototype-based difficulty. In the table, the simple examples are randomly sampled from the simplest 25% of the dataset, while difficult examples come from the hardest 25%. We can see that the simple examples usually refer to common general instructions, while difficult examples tend to be more complex in the language and specify more

| Simple Examples | Difficult Examples |
| --- | --- |
| Can you create an Meeting for Saturday 1 : 00 pm | I had a meeting with Jesse last week |
| Schedule a lunch after the meeting on Thursday. | I need my free day on April 7, to be turned to my All out gallery opening. |
| create a new appointment tomorrow | Add a beer festival in Denver to be for all weekend next week. |

Table 8: Simple and difficult data examples in the SMCalFlow-CS dataset according to the prototype-based difficulty metric. In this table, the simple examples are randomly sampled from the simplest 25% of the dataset, while difficult examples come from the hardest 25% of the dataset.

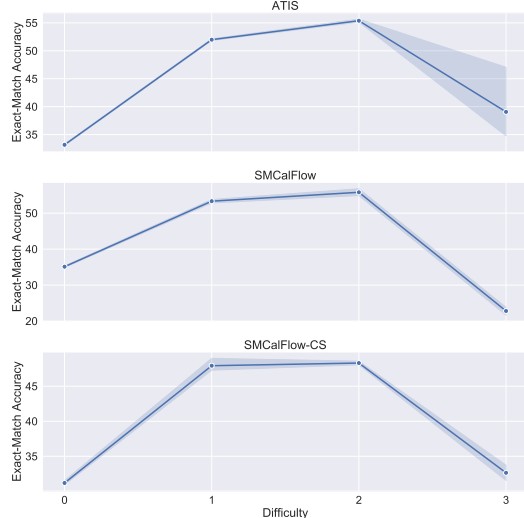

Figure 8: Results on ATIS and SMCalFlow with training sets of different difficulties with learning-based difficulty.

details (e.g., the name Jesse, the place Denver, etc.).

Additionally, we want to point out that while this prototype-based difficulty measure alters the general complexity of the data, it *does not* change the compositional difficulty of the SMCalFlow-CS dataset. This is because the dataset of SMCalFlow-CS is constructed to test the ability to combine multiple instructions in the training set under a few-shot setting (Yin et al., 2021), and the difficulty is controlled by altering the number of few-shot examples. As we always keep the few-shot examples the same in all these training sets and only change the difficulty of the remaining single-skill examples, the clustering step will not introduce other confounding factors to the compositional challenge.

### G.2 Learning-Based Difficulty

We can use an empirical metric based on how soon the model predicts an example correctly during training. This is similar to the popular correctness-based metric (Swayamdipta et al., 2020). Since

the model could reach perfect training accuracy on many of the datasets in this work, we choose to use the dedicated metric of *how soon* the model predicts the example correctly instead of the plain accuracy, so that we can differentiate between examples. Specifically, during training, we evaluate the model on the entire training set for every checkpoint. Then for every example, we log the earliest step when the model has predicted the example correctly for 10 straight checkpoints. See Appendix B for details.

We report additional results of our difficulty-based sub-sampling results in Sec. 5. In Sec 5 in the main paper, we use prototype-based difficulty metrics to conduct the sub-sampling experiments. We make this design choice as if we use the learning-based difficulty metric, then the models perform extremely badly on the most difficult split, which makes it unsuitable for our experiments. The corresponding results are shown in Figure 8.

## H Results with Other Model Configurations

### H.1 Data Augmentation Results with Larger Trained-From-Scratch Transformers

For the main results in the main paper, we use 3-layer Transformers as it is suggested as the best choice for many popular compositional generalization datasets (Csordás et al., 2021). In Table 9, we report the data augmentation number for a larger 6-layer model. We see similar trends on all the datasets, confirming that the advantage brought by increased data complexity is not sensitive to the underlying model scale.

### H.2 Results with Pre-trained Transformers

As the main motivation of this work is to examine how data factors influence model generalization performance, we mainly experiment with models trained from scratch so that we can have clean con-

|  | *Jump* | *Around Right* | GeoQuery (query) | GeoQuery (question) |
|---|---|---|---|---|
| baseline | $4.14_{\pm 4.71}$ | $52.42_{\pm 21.41}$ | $37.63_{\pm 5.57}$ | $58.42_{\pm 1.56}$ |
| 2x Augmentation | $31.79_{\pm 26.23}$ | $79.26_{\pm 7.78}$ | $37.42_{\pm 1.55}$ | $60.57_{\pm 2.96}$ |
| 20x Augmentation | $92.13_{\pm 4.56}$ | $77.50_{\pm 22.34}$ | $41.20_{\pm 2.68}$ | $66.95_{\pm 1.17}$ |
| 200x Augmentation | $99.90_{\pm 0.09}$ | $99.55_{\pm 0.85}$ | $39.68_{\pm 5.08}$ | $62.20_{\pm 1.89}$ |

Table 9: Datasets with increased complexity via data augmentation are easier for compositional generalization. We use 6-layer Transformers and logic form outputs for the GeoQuery dataset in this table.

| Dataset Size | Repetition | GeoQuery (query) | GeoQuery (question) |
|---|---|---|---|
| baseline | / | $67.74_{\pm 5.61}$ | $80.05_{\pm 1.97}$ |
| 20x | / | $72.04_{\pm 10.80}$ | $79.69_{\pm 0.20}$ |
| 20x | Example | $65.41_{\pm 2.54}$ | $81.72_{\pm 1.99}$ |
| 20x | Primitive | $51.08_{\pm 7.27}$ | $61.65_{\pm 4.66}$ |

Table 10: The effect of examples repetition on different datasets for pre-trained Transformers.

trol over all the influencing factors and avoid the interference of pretraining data and testing data. Nonetheless, when we actually deploy models in practical applications, the more common choice is to use pretrained models (e.g., T5 (Raffel et al., 2020), BART (Lewis et al., 2020), etc.). Below we reproduce some of our main results in this paper with T5-base models and summarize the take-aways. For the hyper-parameters, we mostly follow the same setting described in Sec. 2.2. The only change is that we use a learning rate of 0.001 and a linear learning rate scheduler. For the T5 experiments, we report mean and standard deviations from 3 runs.

**Preventing repetition in data is still important.** In Table 10, we reproduce the experiments in Table 4 where we test the effect of repeating examples during the training process. From the results, we can see that similar to the results in the main paper. When the model is repetitively trained on a randomly sampled small set (i.e., the 3rd row in the table), we can only see a small drop in the query split. However, the damage is more substantial when the repeated subset contains few primitives, showing a drop from 72.04 to 51.08 on the query split and a drop from 79.69 to 61.65 on the question split. To summarize, example frequency is still very important during finetuning. Highly repetitive examples can cause substantial damage to the generalization performance, which may also be linked with overfitting on the small subset.

**Pretrained Transformers need less assistance for simple cases.** One crucial difference between the pretrained models and the models trained from scratch is that pretrained models already have a

basic understanding of natural language. For example, by pretraining on a large corpus, it already knows that *run* and *jump* are both verbs; *Texas* and *California* are both location entities, and these words in the same category should be treated similarly. Therefore, it gets a 'jump-start' on these compositional generalization datasets, and may not need a large number of unique primitives to infer the equivalence of *run* and *jump*. As a result, we do not observe consistent improvement by using data augmentation in Table 10.