# OpenReview forum: "Data Factors for Better Compositional Generalization"
_EMNLP/2023/Conference — EMNLP 2023 Main_

### Official Review · Reviewer_3qWv · 2023-08-03

**Soundness:** 4

**Excitement:**

3: Ambivalent: It has merits (e.g., it reports state-of-the-art results, the idea is nice), but there are key weaknesses (e.g., it describes incremental work), and it can significantly benefit from another round of revision. However, I won't object to accepting it if my co-reviewers champion it.

**Paper Topic And Main Contributions:**

This paper aims to analyze (1) the connection between the increased dataset complexity and better compositional generalization; (2) why more complex datasets lead to better generalization. They also give some of the possible reasons as (a) More diverse patterns exemplifid in complex datasets increase the difficulty of surface memorization. (b) Larger dataset size increase the difficulty of simple memorization of samples. (3) How examples of different difficulties inside a dataset affect generalization.

**Questions For The Authors:**

1. As for the medium-level complexity task, would it be any improvement if you first train the model on the easy examples and then gradually increase the difficulty? It'll be more like curriculum learning. The reason that I think the model fails when only training on complex examples might be that there is a gap between the initialized model knowing nothing and a well-trained model knowing complex rules, while using easy examples might close the gap and achive better results than simply training the model on medium-level complexity task.

**Reasons To Accept:**

1. This paper systemetically analyses the effects of the dataset complexity and the scale complexity on the generalization abilities.
2. The idea that only augmenting the dataset with a whole new set of vocabulary while not introducing any new knowledge could boost the performance by a large range is very interesting and intuitively makes sense.
3. The finding that training the model on natrual language datasets with medium level difficulty could achieve the best accuracy is also interesting and might be of interest to the community.

**Reasons To Reject:**

1. The observation that the complexity and large scale would affect the training does not seem to be a very novel observation, it seems to be a consensus in machine learning community.
2. As the difficulty would be hard to measure and the compositional generalization may have not been defined in pretraining, the method and discoveries proposed in this paper do not seem to be very useful for the development of large language models. Generally speaking, we need to choose more diversified and larger pretraining corpus, which is already what they are doing when performing pretraining. I'm kind of curious about what kind of new ideas and inspirations would this paper bring to the NLP community when most of them are working on Large Language Models.

**Reproducibility:**

4: Could mostly reproduce the results, but there may be some variation because of sample variance or minor variations in their interpretation of the protocol or method.

**Reviewer Confidence:**

3: Pretty sure, but there's a chance I missed something. Although I have a good feel for this area in general, I did not carefully check the paper's details, e.g., the math, experimental design, or novelty.

---

> ### Author Rebuttal · Authors · 2023-08-29
>
> Thanks to the reviewer for appreciating our systematic analysis and our AugZero idea. We also thank the reviewer for raising the important point on novelty and the connection to LLMs.
>
> **On the novelty of the data complexity analysis**
>
> We agree with the reviewer that other papers have studied the broad topic of complexity and observed the effect of large-scale datasets, as we acknowledge at the very beginning of our work at LINE 056-058. However, to the best of our knowledge, there are still very few works explaining the reason behind the empirical gain. Our work is motivated by such mysterious benefits, and we explore hypotheses explaining this benefit via the angle of repetition frequency (Sec. 3). Our hypothesis also leads to a novel data augmentation algorithm, AugZero, which is acknowledged by Reviewer 9qDf.
>
> **On implications on LLMs**
>
> In Sec. 7, we provide a discussion about the implications of findings in our work on LLMs. Our connection is more on the finetuning stage of LLMs instead of pretraining, which is how most people use LLMs due to computational constraints. Indeed when we do pretraining, the common practice is to choose a more diversified and larger corpus. However, this is not always possible for finetuning. To give a more concrete example, during finetuning, we sometimes only have access to a small dataset with limited diversity. In this case, we will face the choice to either finetune the model directly on the small subset, or do some augmentation to diversify the small subset. The results in our work indicate that the latter may be a better choice (see LINE 618-637). We will further clarify these implications in our final version in Sec. 7. In addition, please also check our results using pretrained Transformers in Appendix F.2 showing the generalizability of our findings.
>
> **Response to Question 1**
>
> Thank you for the suggestion! Training the model with an easy-to-difficult curriculum is a great idea. Its benefit is also suggested in the Mix results in Table 4. Following your suggestion, we trained the model with an easy to difficult curriculum on SMCalFlow using the data in Figure 4 and the same total training steps. The overall model achieves an accuracy of 50.35, substantially higher than the model only trained on the hardest subset. While these two performances are not directly comparable as more data are used in the curriculum training model, it proves the benefit of training first on easy data before training on difficult data. To further find the best curriculum for generalization is a promising direction and we will explore it in our future works.

---

### Official Review · Reviewer_CqGi · 2023-08-05

**Soundness:** 3

**Excitement:**

3: Ambivalent: It has merits (e.g., it reports state-of-the-art results, the idea is nice), but there are key weaknesses (e.g., it describes incremental work), and it can significantly benefit from another round of revision. However, I won't object to accepting it if my co-reviewers champion it.

**Paper Topic And Main Contributions:**

This paper explores an interesting issue with several NLP tasks: a model trained on a larger dataset or a more challenging task can generalise better on these specific tasks, compared to training the same model from scratch. To understand this inconsistency, the authors show that a model trained on a larger or more complex dataset performs better on the same task.

**Questions For The Authors:**

Please refer to the above section.

**Reasons To Accept:**

This paper provides interesting insights into why models perform better on larger and more complex data. The method employed seems to be sound (up to a point, please refer below).

**Reasons To Reject:**

One major issue I see in this study is the fact that the *computational* factor seems to be largely ignored. In particular, in lines 448-449, the authors state that "For this section, we report results with three runs as the dataset sizes are substantially larger". Is this factor not giving an  unfair advantage to a model trained on a larger dataset? Should a fair comparison not use the same amount of compute in both setting (e.g. smaller vs larger dataset)?

**Reproducibility:**

4: Could mostly reproduce the results, but there may be some variation because of sample variance or minor variations in their interpretation of the protocol or method.

**Reviewer Confidence:**

4: Quite sure. I tried to check the important points carefully. It's unlikely, though conceivable, that I missed something that should affect my ratings.

---

> ### Author Rebuttal · Authors · 2023-08-29
>
> Thank you for your helpful comments and we are glad you acknowledged the insight provided by our experiments.
>
> **On the importance of the computation factor**
>
> Thank the reviewer for highlighting the computational factor and we agree on its importance. However, our experiment design has already taken that factor into consideration. For any experiments in the same table, we always train the models using the same amount of total steps even though they are trained on datasets of different sizes (e.g., 2*, 20*, 200* Augmentation in Table 1). Hence, the comparison in every table is always under the assumption of using the same computation cost. We report some related details at LINE 382-383 and LINE 1112-1115. We decided to run fewer runs for large datasets mainly because of data preprocessing overhead and the slow convergence on large datasets, but we give the same computation budget to every model and train them till convergence. We will further clarify this point in our final version.

---

### Official Review · Reviewer_P8xF · 2023-08-10

**Soundness:** 3

**Excitement:**

3: Ambivalent: It has merits (e.g., it reports state-of-the-art results, the idea is nice), but there are key weaknesses (e.g., it describes incremental work), and it can significantly benefit from another round of revision. However, I won't object to accepting it if my co-reviewers champion it.

**Missing References:**

- Example level complexity: Bogin et al. Unobserved local structures make compositional generalization hard

**Paper Topic And Main Contributions:**

This paper provides a diagnostic and empirical study of data factors that will influence the compositional generalization of Transformer models trained from scratch. In particular, it studies dataset scale and pattern complexity and finds that increasing dataset complexity can improve the compositional generalization of models. It offers further hypotheses and analysis about why increased complexity would improve generalization. Finally, the paper explores how example difficulty influences generalization differently in both synthetic and realistic datasets.

**Questions For The Authors:**

A. Have the authors considered and explored other measurements of data complexity, e.g., compound divergence (Keysers et al., Shaw et al.), local structures (Bogin et al.)? If so, how do they influence generalization performance?

B. Have the authors studied other compositional splits for SCAN (e.g., MCD from Keysers et al.) and GeoQuery (e.g., TMCD from Shaw et al.)? And other semantic parsing datasets that evaluate compositional generalization, e.g., CFQ (see a similar study from Tsarkov et al.)?

C. Can the authors clarify the differences between “improving compositional understanding” and “preventing surface memorization”. And why do we need to distinguish them and explicitly characterize these two model behaviors? Aren’t these two factors always correlated? The discussions in L288-L310 could benefit from further clarification.

D. SCAN* creation details: L1074-1078, is there any specific reason for this design? Based on my experience, one of the most significant challenges of SCAN is separating the “around” and “opposite”.

E. Can the authors clarify more how the section starting from L378 supports the two proposed hypotheses in section 4.2? Theoretically, it’s still possible to have large datasets with example repetition or primitive repetition curriculum (bottom of Table 3)? The experimental setup used here does not seem to mimic the standard training practice. I wonder how much of the conclusion drawn here can directly support the claim that reducing frequently recurring examples improves compositional generalization.

F. Is there any qualitative analysis of the difficulty metric, especially for the prototype-based metric in the context of a natural language dataset? How effective is it at clustering examples? I imagine that for some compositional splits, such as SMCalFlow-CS, the "hard" examples might be quite similar in the embedding space. For instance, training examples like "who is my manager" and "schedule a meeting with Amy," and a test example like "schedule a meeting with my manager," could be very close in the embedding space. Therefore, this measure might not be a good indicator of example-level difficulty. Other measurements, such as those by Bogin et al. (see below), might provide more informative signals.

G. Figure 4: IIUC, difficulty here refers to the L2 distance to the centroid. If so, why does it appear to always be an integer?

**Reasons To Accept:**

- This paper provides diagnostic studies on data factors that influence compositional generalization, which could be helpful for understanding the generalization performance of models by isolating any factors from pre-training.
- Experiments and analysis are well-designed and motivated from prior work. The data and experiments are well controlled.

**Reasons To Reject:**

- The main experiments on data scale and pattern complexity only use relatively easier splits from SCAN and GeoQuery. Many other more challenging compositional splits have been proposed but were not considered: for SCAN, MCD splits (Keysers et al., Shaw et al.); for GeoQuery, TMCD splits (Shaw et al.). I'm not sure if this conclusion will generalize to other datasets or even other splits, especially since the pattern complexity is only based on the number of primitives without taking into account other complex structures.
- While the experiments on example difficulty consider more realistic datasets, the complexity metric is not very convincing. There is no evidence to support whether the prototype-based difficulty metric is reliable or not.

**Reproducibility:**

4: Could mostly reproduce the results, but there may be some variation because of sample variance or minor variations in their interpretation of the protocol or method.

**Reviewer Confidence:**

4: Quite sure. I tried to check the important points carefully. It's unlikely, though conceivable, that I missed something that should affect my ratings.

**Typos Grammar Style And Presentation Improvements:**

- Table 2: The legend could be clearer. For example, it could directly define "+Both" as "20x augmentation + more primitives."
- Good to have dataset statistics and examples in the appendix.
- L1132: speed-based difficulty metric is not mentioned in the main text.
- Section 5 seems not to be closely related to Sections 3-4, and even uses some different datasets. It might be helpful to clarify these points in the main experimental setup, and perhaps consider making Section 4 a subsection of Section 3.

---

> ### Author Rebuttal · Authors · 2023-08-29
>
> Thank you for your helpful comments and we are glad you appreciate our findings in isolating different data factors and how we control them in experiments.
>
> **Response to R1 and Questions A, B**
>
> The motivation of our work is to study how larger and more complicated datasets make composition easier to learn. Therefore, we conduct our experiments by precisely controlling the dataset complexities (e.g., number of primitives, length) in order to find how it affects the resulting model’s compositional generalization. We are able to control the number of unique primitives following the previous prim2primX data augmentation and control the length by simply cutting off the training set at different lengths threshold. The complexity of MCD-based splits, however, does not change significantly when the dataset becomes larger. One way to test a related hypothesis connected to our work is to check if changing the number of unique local structures can improve performance on MCD-based datasets. However, controlling the number of unique local structures is very difficult and we are not aware of any existing work that managed to automatically create new structures. Therefore, we mainly focus on primitive generalization and length extrapolation in this work (with more realistic datasets on example difficulty experiments), and leave the exploration on more complex structures for future work.
>
> **Response to Question C**
>
> Indeed, “improving compositional understanding” and “preventing surface memorization” are always correlated. As we stated in LINE 288-310, the model’s behavior is a complicated combination of these two competing behaviors. A method that improves compositional generalization might (1) directly encourage compositionality through certain inductive bias (e.g., a neural-symbolic model), (2) or it does not provide such an inductive bias but instead punishes the model for surface memorization. Increasing pattern and scale complexity belong to the second category as they increase the cost of memorization without increasing the cost of generalization. We believe this is an important distinction to make as we are trying to understand the source of improvement in Sec. 4.2 and Sec. 4.3.
>
> **Response to Question D**
>
> Thank you for pointing out this issue. We apply this simplification by accident. However, this simplification will not influence our findings in Table 1 and Table 2. All our models in the comparison are trained and tested using datasets with the same syntax. Additionally, in all our experiments related to SCAN* around right or the original SCAN around right (Table 1 and Table 2), we never noticed any difficulty understanding the around and opposite parts.
>
> **Response to Question E**
>
> Here we explain the function of the paragraph starting at LINE 378 in Sec. 4. The overall target of Sec. 4 is to explain the benefit of increased dataset complexity. One of the hypotheses we proposed in Sec. 4 is that increasing dataset complexity can help compositional generalization by reducing frequently recurring examples (LINE 337-339). Note that we do not claim that increasing dataset complexity will always lead to reducing frequently recurring examples (as shown by our examples at the bottom of Table 3), however, this is mostly true in cases where the examples are collected naturally. This hypothesis is jointly supported by two sets of observations: (1) naturally increasing the dataset size helps generalization, and (2) naturally increasing the dataset size while still adding highly recurring examples during training will lead to worse results. The paragraph starting at LINE 378 shows observation (2) while Section 3 shows observation (1). We will further clarify this in the final version.
>
>
> **Response to R2 and Question F**
>
> As we pointed out in the paper, we designed the prototype-based difficulty metric based on [1], where they showed that in ImageNet classification, pruning the data using the prototype-based metric matches or exceeds the performance of pruning with the best supervised difficulty metric, memorization. The memorization-based difficulty metric is very costly since it is computed as how much the probability of predicting the correct label for the example increases when it is present in the training set relative to when it is absent [1]. The fact that the prototype-based metric matches its performance in data pruning shows it is a reliable metric. On the SMCalFlow data, the clustering results are also reasonable. We see more standard queries like “Put on a meeting on thursday” in the easiest split, as there are many similar queries just changing their day, while in the hardest split, queries contain more specific information like “Can you please add an obligatory family visit on my calendar on wednesday next week at 6pm”. We will expand this discussion in the final version, provide clustering result examples to better demonstrate its validity, and mention other related complexity metrics.
>
> More specifically for our experiments on SMCalFlow-CS, clustering metrics will be reliable and will not bring confounding problems as in Question F. This is because the dataset of SMCalFlow-CS is constructed to test the ability to combine multiple instructions in the training set (See the SMCalFlow Compositional Skills paragraph in Yin et al., 2021) under a few-shot setting. As we always keep the few-shot examples the same in all these training sets and only change the difficulty of the remaining single-skill examples, the clustering step will not introduce other confounding factors to the compositional challenge. We will clarify this point in the final version.
>
>
> **Response to Question G**
>
> In Figure 4, the x-axis actually shows the quantile of the difficult examples based on the distance to the centroids. Therefore, “0” means examples in the easiest quantile while “3” means examples in the most difficult quantile. We will change the x-axis title to clarify this.
>
> [1] Beyond neural scaling laws: beating power law scaling via data pruning. NeurIPS 2022.

---

### Official Review · Reviewer_9qDf · 2023-08-10

**Soundness:** 3

**Excitement:**

4: Strong: This paper deepens the understanding of some phenomenon or lowers the barriers to an existing research direction.

**Paper Topic And Main Contributions:**

The paper provides an empirical study of compositional generalization in seq2seq models, trying to tease apart the factors that promote a model's ability to generalize compositionally. This question is timely as it is inserted in the broader research goal of elucidating the training factors that allow LLMs pretrained on large datasets to acquire some of their reasoning and generalization abilities, even though they were not explicitly trained with the intent of displaying those abilities.
The results are interesting as they identify dimensions characterizing dataset complexity that contribute to promoting generalization performance in compositional seq2seq tasks that are actionable to the point of suggesting a data augmentation procedure that is also demonstrated in the paper.

### Post-review discussion changes
Following the authors' rebuttal I increased my score:
* "Soundness" was increased to "good"
* "Excitement" was increased to "strong"

**Questions For The Authors:**

- If preventing memorization improves compositional generalization, then regularization or other techniques to prevent memorization (overfitting) should also help. Is there any indication that this might be the case, or is that a completely wrong supposition?
- The idea behind the AugZero data augmentation procedure is rather neat. It seems that there might be an additional "combinatorial factor" that could be exploited in terms of increasing the dataset by mixing tokens across tokenizers. For instance, the sequence "walk left and jump" is augmented by AugZero to "walk2 left2 and2 jump2" using an additional tokenizer. But in principle the sequence "walk left2 and2 jump2" obtained by using "walk" from the first tokenizer and the remaining tokens from the second one could also be used as a data augmented sample. Would that indeed be a valid augmentation? And if so, would there be any advantage to exploiting the resulting combinatorial factor in terms of combining tokenizers and tokens replacement?

**Reasons To Accept:**

- The paper identifies a set of factors characterizing a dataset that are positively correlated with compositional generalization performance by influencing a trade off between memorization and generalization.
- the paper proposes a novel "zero knowledge" data augmentation procedure motivated by this finding.
- The paper is well structured as it is presented as an hypothesis-driven empirical study.

**Reasons To Reject:**

- The paper has an understandable but rather idiosyncratic focus on a specific instantiation of compositional generalization embodied by the SCAN dataset where the main results are obtained
- The paper characterizes some of the data factors that underlie compositional generalization in a strictly phenomenological fashion, without relating them to the underlying mechanism in terms of statistical learning theoretic quantities. In particular, the presumed "memorization" factor being tweaked by the dataset complexity seems clearly related to overfitting, suggesting that the crossover from memorization to compositional generalization is not merely dataset-dependent, but also has to do with the model's capacity and the inductive bias deriving from the model architecture. This type of consideration are glaringly missing in the paper.
- Another aspect that is left oddly unexplored by the paper are mechanisms underlying compositional generalization that are understood in terms of representation learning. In the end the algorithm being analyzed is a machine learning model meaning that, mechanistically, compositional generalization has to come about due to some geometric property of the embedding representing the sequences constituting the task. Moreover, these mechanisms can be potentially directly investigated by examining the model's activation. However, also in this regard the paper disregards this opportunity to access, examine and mechanistically understand the phenomenon of compositional generalization, deciding to limit the investigation to a phenomenological investigation.

**Reproducibility:**

4: Could mostly reproduce the results, but there may be some variation because of sample variance or minor variations in their interpretation of the protocol or method.

**Reviewer Confidence:**

4: Quite sure. I tried to check the important points carefully. It's unlikely, though conceivable, that I missed something that should affect my ratings.

---

> ### Author Rebuttal · Authors · 2023-08-29
>
> Thank you for your helpful comments and we are glad that you appreciate our study as well-structured,  you like our novel AugZero data augmentation method, as well as our findings on the relationship between data factors and compositional generalization.
>
>
> **On datasets used in this work**
>
> We use SCAN as one of the main datasets used in this work as its well-controlled syntax makes it easier to design clean experiments supporting our hypotheses. It is also the major dataset used in Patel et al., 2022 and Jiang et al., 2022, the two papers motivating this study. In addition to the SCAN dataset, to demonstrate the generalizability of our findings, we have also tested our findings on three other natural language datasets: GeoQuery, SMCalFlow, and ATIS.
>
>
> **Other non-data factors that affect compositional generalization**
>
> We acknowledge that compositional generalization can be affected by other non-data factors, including model capacity [1] and model architecture [2]. Those other factors are all crucial but are also popular topics studied by previous work (see LINE 047-049, LINE 568-584, and also [1,2]). In this work, we decide to conduct a focused, systematic study on various under-explored data factors that affect compositionality using a commonly used architecture (seq2seq transformer) and a fixed model capacity.
>
> **Representation geometry**
>
> We thank the reviewers for the suggestion on understanding compositionality in terms of representation learning. This is definitely an important direction as shown in past works [4]. However, those works have different motivations from our work. Geometry-based study helps us better understand *what* is the composition learned inside the models, while our work focuses more on *why* models learn better composition on more complex data. These two directions are complementary to each other. To demonstrate this, below we provide a simple example of how we can explore certain geometry of learned representations to analyze why the model trained on more complex datasets achieves stronger generalization:
>
> We calculate the cosine-similarity between the embedding of “jump” and “walk” learned from the original SCAN jump split or the “prim2primX” augmented data. When the model is trained on the original SCAN jump split, the similarity between the embeddings of “jump” and “walk” is very low, at 0.34. When the model is trained on the prim2primX-augmented SCAN jump, the similarity increases to 0.88. This suggests that when the model is trained on a dataset with higher pattern complexity, the model can better capture the syntactic similarities between different primitives. This leads to stronger primitive-level compositional generalization because when the model has learned the similarity between “jump” and “walk”, it is more likely to understand an unseen sentence “jump twice” by inferring from the seen sentence “walk twice”. In our final version, we will provide a discussion about the connection between different directions on the interpretability of compositionality.
>
>
> **Response to Question 1**
>
> There are indeed works that show various regularization techniques help compositional generalization. Simple L2 regularization is used in [2] and more advanced regularizations are proposed in [3]. We’d also like to point out that overfitting is a broad umbrella term that covers many different phenomena. Even with proper regularization, most models achieve perfect accuracy and “overfit” on the training set, so figuring out which regularization works is not trivial. Furthermore, recent advanced LLMs (e.g., GPT4) demonstrate significantly stronger performance on datasets like SCAN. These LLMs can automatically acquire compositional behavior without explicitly designed regularization. Therefore, we are interested in what aspects of the LLM training procedure provide effective implicit regularization. Our study provides related insights (i.e., the benefit of training on complex data) in this direction.
>
> **Response to Question 2**
>
> Thanks for the suggestion! "Walk left2 and2 jump2" is indeed a valid augmentation. In fact, the “prim2primX” data augmentation used on SCAN in Sec. 3 and 4 includes very similar examples (See Appendix C.1 for details). The only difference is that we do not augment functional words like conjunctions (“and2”). As shown in Table 2 (“+ Both”). From Table 2, we can see that prim2primX outperforms AugZero, partly due to having these additional "combinatorial factor" results. However, doing such augmentation requires inducing the “walk -> WALK” mapping beforehand, which is not trivial on natural language datasets (even for simple ones like GeoQuery). On the contrary, AugZero does not require any syntax induction and has the advantage of simplicity.  Additionally, we also explored another related idea that randomly using walk, walk1, walk2, … in all the examples, which make the one-to-one map “walk->WALK” becoming a many-to-many map “walk, walk1, walk2, …->WALK, WALK1, WALK2, …”, but do not observe further improvements. We will clarify these variants in our final version.
>
> [1] Evaluating the Impact of Model Scale for Compositional Generalization in Semantic Parsing. EMNLP 2022
> [2] Compositional Generalization for Primitive Substitutions. EMNLP 2019
> [3] Consistency Regularization Training for Compositional Generalization. ACL 2023
> [4] Characterizing Intrinsic Compositionality in Transformers with Tree Projections. ICLR 2023

---

### Meta-Review · Area_Chair_TruK · 2023-09-14

**Recommendation:** 3

**Metareview:**

This work investigates compositional generalization in seq2seq models, and aims to disentangle the factors that enable a model to generalize compositionally.

The submission received 4 reviews. Overall, the reviewers agree that the soundness of the proposed work is sufficient to support its main arguments, though all point out additional points that should be addressed in a revision. These points include adding further motivation for focusing on the specific tasks and datasets and a justification for the complexity metric used, and improving the clarity of several sections. While the work is judged to be largely sound and a good contribution, it is less clear how exciting the resulting insights will be to the EMNLP audience.

---

### Decision · Program_Chairs · 2023-10-07

**Decision:**

Accept-Main

**Comment:**

This work investigates compositional generalization in seq2seq models, and aims to disentangle the factors that enable a model to generalize compositionally.

The submission received 4 reviews. Overall, the reviewers agree that the soundness of the proposed work is sufficient to support its main arguments, though all point out additional points that should be addressed in a revision. These points include adding further motivation for focusing on the specific tasks and datasets and a justification for the complexity metric used, and improving the clarity of several sections. While the work is judged to be largely sound and a good contribution, it is less clear how exciting the resulting insights will be to the EMNLP audience.